# Bull Shark (*Carcharhinus leucas*) Occurrence along Beaches of South-Eastern Australia: Understanding Where, When and Why

**DOI:** 10.3390/biology12091189

**Published:** 2023-08-31

**Authors:** Amy F. Smoothey, Yuri Niella, Craig Brand, Victor M. Peddemors, Paul A. Butcher

**Affiliations:** 1NSW Department of Primary Industries, Fisheries Research, Sydney Institute of Marine Science, Mosman, NSW 2088, Australia; vic.peddemors@dpi.nsw.gov.au; 2Department of Biological Sciences, Macquarie University, North Ryde, Sydney, NSW 2113, Australia; yuri.niella@mq.edu.au; 3NSW Department of Primary Industries, Fisheries Research, National Marine Science Centre, Coffs Harbour, NSW 2450, Australia; craig.brand@dpi.nsw.gov.au (C.B.); paul.butcher@dpi.nsw.gov.au (P.A.B.)

**Keywords:** acoustic telemetry, intra-specific variability, animal movement, seasonality, spatial, temporal, shark bite mitigation

## Abstract

**Simple Summary:**

In Australia, bull sharks are one of the species implicated most regularly in shark–human interactions. The factors affecting bull shark presence in nearshore waters of New South Wales (NSW), Australia were examined to determine periods of increased overlap with beach-users and thereby potentially increased risk of shark–human interactions. We investigated the spatial ecology of 233 juvenile and large (including sub-adult and adult, >175 cm total length) bull sharks acoustically tagged and monitored over a 5.5-year period (2017–2023) along 21 coastal beaches of NSW. Our study highlights that large bull sharks were present more in waters north of 32° S with a southward distribution during summer and autumn. The occurrence of large bull sharks in nearshore waters was greatest from midday to 04:00, when water temperatures were higher than 20 °C, after >45 mm of rain and when swell heights were between 1.8 and 2.8 m. We show that current shark bite mitigation educational messaging, incorporating proximity to rivers, turbidity (rainfall) and time of day, reflect heightened periods of the occurrence of large bull sharks. We concur with the current shark smart advice that nocturnal swimming and surfing, especially in warm waters and when water visibility is poor, should be avoided for many reasons, not the least of which being the potential presence of bull sharks. However, we suggest that time of day messaging for large bull shark presence should be modified from “dawn and dusk” to instead refer to afternoon and low-light periods.

**Abstract:**

Unprovoked shark bites have increased over the last three decades, yet they are still relatively rare. Bull sharks are globally distributed throughout rivers, estuaries, nearshore areas and continental shelf waters, and are capable of making long distance movements between tropical and temperate regions. As this species is implicated in shark bites throughout their range, knowledge of the environmental drivers of bull shark movements are important for better predicting the likelihood of their occurrence at ocean beaches and potentially assist in reducing shark bites. Using the largest dataset of acoustically tagged bull sharks in the world, we examined the spatial ecology of 233 juvenile and large (including sub-adult and adult) bull sharks acoustically tagged and monitored over a 5.5-year period (2017–2023) using an array of real-time acoustic listening stations off 21 beaches along the coast of New South Wales, Australia. Bull sharks were detected more in coastal areas of northern NSW (<32° S) but they travelled southwards during the austral summer and autumn. Juveniles were not detected on shark listening stations until they reached 157 cm and stayed north of 31.98° S (Old Bar). Intra-specific diel patterns of occurrence were observed, with juveniles exhibiting higher nearshore presence between 20:00 and 03:00, whilst the presence of large sharks was greatest from midday through to 04:00. The results of generalised additive models revealed that large sharks were more often found when water temperatures were higher than 20 °C, after >45 mm of rain and when swell heights were between 1.8 and 2.8 m. Understanding the influence that environmental variables have on the occurrence of bull sharks in the coastal areas of NSW will facilitate better education and could drive shark smart behaviour amongst coastal water users.

## 1. Introduction

Mitigating human–shark interactions has become a complex issue for management agencies, particularly given the increased frequency and temporal and spatial clusters that have occurred around the world [1,2]. The increase in shark bite incidents has been attributed to the growth in the human population coupled with an increase in the number of people participating in water-based activities [1,2,3]. These factors, coupled with changes in prey availability, sea surface temperature, rainfall, distance to river mouths, geographical location, fishing activities and changes in shark patterns of occurrence and behaviour [4,5], have also been suggested to influence the risk of shark bites. Despite the incredibly low probability of a shark bite occurring [6], when they occur, they receive large amounts of negative media attention [1,5,7], escalating fear from the public and resulting in demands for local government agencies to enhance shark mitigation strategies to increase safety and awareness.

Shark mitigation programs have historically focused on catch-and-kill mitigation measures targeted at bull (*Carcharhinus leucas*), tiger (*Galeocerdo cuvier*), and white (*Carcharodon carcharias*) sharks [8,9,10], the species responsible for the majority of serious and fatal interactions. However, concerns associated with the impacts of lethal mitigation measures on marine wildlife has recently led to shifts in priorities towards non-lethal mitigation strategies (drones [11,12]; SMART drumlines [13,14]; personal shark deterrent devices [15,16,17]) and increasing our understanding of the ecology of bull, tiger and white sharks [18,19,20,21,22,23,24,25,26,27]. Knowledge of their occurrence and movement behaviour could enhance the predictability of shark encounters and thereby potentially reduce the risk of shark bites through advising beach authorities and the public to modify human behaviour in areas, times and conditions of increased risk, coupled with deploying site-specific mitigation measures to reduce the risks of negative human–shark interactions.

Along the east coast of Australia, a seasonal increase in shark interactions (and beach usage by humans) occurs between November and April with most serious bites attributed to white, bull and tiger sharks [2,3,5,28]. During the spring and summer of 2014/15 and 2015/16, a cluster of unprovoked shark interactions occurred in New South Wales (NSW) leading to the NSW government establishing the NSW Shark Management Strategy (SMS). The aim of the SMS was to trial a suite of shark bite mitigation measures to increase protection for water-users while minimising harm to target (white, tiger and bull sharks) and non-target species [29]. One mitigation measure trialled was the deployment of 21 satellite-linked listening stations (VR4-Global, Innovasea; [30]) along the NSW coast to detect acoustically tagged target sharks and enable real-time public alerts about them off those beaches.

Bull sharks (*Carcharhinus leucas*) are distributed throughout rivers, estuaries, nearshore and continental shelf waters of tropical, sub-tropical and warm temperate regions [31,32]. In Australia, the reported distribution of the species is across the northern half of the continent from Sussex Inlet (35.16171° S; [20]) on the east coast, to Perth (31.9523° S) on the west coast (Figure 1A; [33]), with male-driven genetic connectivity and female philopatric behaviour throughout its range [34]. Studies carried out to investigate the movement patterns and drivers of bull sharks along the east coast of Australia have shown that their behaviours are complex, vary according to the size and sex of individuals [35,36,37,38] and across different spatial and temporal scales [18,19,39,40,41]. Sub-adult and adult bull sharks migrate large distances and exhibit strong site fidelity on a seasonal basis to estuaries and coastal areas [18,20,40]. Their movements have been linked to the availability of wetlands and estuarine habitats [42], localised heavy rainfall [38,39] and warm water temperatures [18,19,20,41]. While recent studies have investigated the drivers of movement patterns along the east coast of Australia, our study is the first to report on the long-term patterns of the occurrence and seasonal trends in the movements and environmental drivers of juvenile and large (sub-adult and adult) bull sharks over an 8° latitudinal span of coastal beaches. Here, we report on the spatial ecology of 233 bull sharks acoustically tagged and monitored over a 5.5-year period (2017–2023) along the coast of New South Wales, Australia, specifically to: (i) determine the seasonal and diel patterns in the nearshore occurrence of bull sharks, and (ii) test whether these patterns of occurrence are influenced by environmental and biological drivers. The broader research findings presented here will provide essential knowledge of large bull shark behaviour that will ultimately enhance predictions of bull shark occurrence in coastal areas of NSW, thereby contributing to beach-user education and their assessment of the risk of shark bites.

## 2. Materials and Methods

### 2.1. Shark Tagging and Acoustic Telemetry Data

A total of 233 bull sharks (female = 120 and male = 113) were tagged between 9 March 2009 and 14 December 2022 with V16-6L acoustic transmitters (Innovasea) programmed on a pseudo-random repeat rate of 30–90 s (*n* = 2) or 40–80 s (*n* = 231) with a battery life of 10 years. Over this time, 7 juvenile and 2 adult tagged bull sharks are known to have died following capture by fishers. Since the dates of the capture by fishers for 6 of these juveniles and 1 of the adults was unknown, these 7 sharks were excluded from all subsequent analyses. Tagging was carried out in estuarine and coastal waters of NSW (Figure 1A) with most sharks caught using bottom-set longlines deployed in estuaries (*n* = 131, hereafter referred to as set-lines, as described in Smoothey et al. [43], Appendix A), while the rest were caught using (i) Shark-Management-Alert-in-Real-Time (SMART) drumlines (*n* = 84 [13,14,44]), or (ii) rod and reel (*n* = 18). Sharks captured using set-lines and SMART drumlines were brought alongside the vessel and secured with cross-pectoral and tail ropes. Sharks caught by rod and reel were brought to shore, secured by a tail rope and restrained in water depths sufficient to completely cover their gills. A total of 90 individuals were fitted with external transmitters embedded using nylon umbrella or stainless-steel anchors, 3–6 cm into the musculature below the first dorsal fin with applicator needles mounted on an aluminium hand-shaft. Another 143 sharks were internally tagged, where a small (3 cm) incision was made in the ventral midline to fit the transmitter into the peritoneal cavity and the wound closed using two interrupted sutures (see [18]). All captured individuals were tagged with an external identification tag below the first dorsal fin. Surgical procedures were performed following protocols approved by NSW Department of Primary Industries Fisheries Animal Care and Ethics (permit 07/08) following veterinary training of staff. All sharks were sexed and their total length (TL) measured to the nearest cm.

Tagged sharks were monitored by 21 satellite or cellular-linked acoustic receivers (Innovasea, historically known as VR4Gs, hereafter referred to as shark listening stations [30]) positioned at 21 locations along the NSW coast (Figure 1A). Shark listening stations were deployed 500 m from shore in 6–16 m, with hydrophones set 4 m below the surface (for more details on the design of the array see Spaet et al. [26]). Range testing revealed that the detection range of the shark listening stations was, on average, between 200 and 500 m [26]. All receivers were checked daily, serviced annually and operated continuously between December 2015 and January 2023, except for the receivers at Kingscliff, Evans Head, Yamba, Port Macquarie, Kiama and Merimbula (Figure 1A), which were not operational between 7 and 23 days between July and December 2018 due to technical issues. All receiver non-operational days were subsequently excluded from the analyses.

### 2.2. Environmental Data

A series of environmental variables were tested for their possible influence on the occurrence of bull sharks along the coast of NSW (Table 1). Location of river mouths were downloaded from the Australian Department of Agriculture (available at https://www.environment.gov.au/fed/catalog/search/resource/details.page?uuid=a8bd8d06-813b-4420-908b-a79c065cf533; accessed on 26 April 2023) and used to calculate the respective distances from each shark listening station using the geosphere R package [45]. Hourly wave data for swell height and tidal height were obtained from the Manly Hydraulics database (available at https://mhl.nsw.gov.au/; accessed on 26 April 2023) from the nearest stations to each shark listening station. Daily rainfall data were downloaded from the Australian Bureau of Meteorology (http://www.bom.gov.au/climate/data/; accessed on 26 April 2023), and also from the nearest meteorological stations to each shark listening station. Water temperatures were obtained from sentinel tags deployed on the mooring line of each shark listening station and lunar phase calculated using the lunar R package, respectively [46].

### 2.3. Statistical Analyses

Quality control was applied to the raw acoustic data by identifying and removing any false detections prior to analysis [47]. The raw detections of bull sharks were then weighted against the detection probability of the sentinel tags on each shark listening station, found to be significantly higher between 4 a.m. and midday but not as a function of station latitude (Appendix A), in order to remove any further false detections in the dataset (i.e., single detections during periods of significantly lower detection probability). Since acoustic telemetry data only provide information on the presence of bull sharks detected by each shark listening station, a standardisation was performed to include absence data. For this purpose, the total monitoring period was divided into binned hourly intervals for each day of tracking. Model suitability was evaluated using k-fold cross validation.

To assist with future recommendations for researchers and managers of programs incorporating acoustic tagging of bull sharks, we investigated the effect of tag position on the longevity of detections. This was carried out by comparing the frequency of detections and total length of detected time for externally tagged sharks with those internally tagged and investigated subsequent detections of bull sharks beyond the shark listening stations. Additional detection data were downloaded from the Integrated Marine Observing System Animal Tracking Database (available at https://animaltracking.aodn.org.au; accessed on 4 July 2023).

To investigate intra-specific variation in the occurrence of bull sharks, tracked individuals were divided into two biological groups according to previous studies carried out on juvenile foraging patterns [48] and sex-specific maturation sizes reported by Cruz-Martinez et al. [49]. Juvenile bull sharks are known to remain inside of their natal estuaries for several years after birth [36], and in the north coast of NSW they were found to make greater use of coastal waters when they become larger than 175 cm TL [20,48]. We subsequently refer to juveniles as male and female bull sharks smaller or equal to 175 cm TL. Size-specific groups included sub-adult and adult sharks larger than 175 cm TL at tagging, referred to as large sharks, or juvenile sharks likely to have grown larger than the 175 cm TL at the time of detection (based on the sex-specific von Bertalanffy growth equations from Cruz-Martinez et al. [49]). The presence/absence of each biological group was then matched to this standardised dataset using the acoustic detection data in which presence was attributed to each shark listening station that recorded at least one individual within a day and hour. All analyses were carried out using the R software (version 4.1.1).

As detections of juveniles only constituted 15% of the overall data and this size category is unlikely to be implicated in shark bites [3], environmental factors influencing the nearshore occurrence of large bull sharks were investigated using only animals >175 cm (including juveniles that had reached this size at time of detection) via generalised additive mixed models (GAMM), with the mgcv R package [50]. Variation in the data, expressed as error, was explored using a family of binomial probability distribution functions (Table 1). Year was included as a random factor in all GAMMs to account for possible inter-annual variation in the trends observed. In addition, a second model quantified spatial and temporal variation in the intra-specific patterns (Table 1) of bull shark occurrence along the coast of NSW. This generalised additive model (GAM) also used a binomial family of errors and tested interactions between the variables latitude and year, to quantify spatial trends varying across temporal scales, and hour of the day (Table 1). New significant variables were gradually added to a previous simpler nested model according to their higher Akaike Information Criterion (AIC) weights. Collinearity among predictor variables was assessed with Pearson’s correlations. To account for the number of sharks tagged potentially influencing the likelihood of detections, and therefore their presence along the study area, the logarithm of cumulative total number of sharks tagged was included as an offset covariate in all models. The final models were inspected for a normal residual distribution.

## 3. Results

The tagged individuals comprised 89 large females (minimum size = 175 cm, mean = 236.5 cm, maximum = 322 cm TL), 91 large males (minimum = 176 cm, mean = 248.3 cm, maximum = 312 cm TL), 31 juvenile females (minimum = 81 cm, mean = 136.8 cm, maximum = 174 cm TL) and 22 juvenile males (minimum = 80 cm, mean = 131.9 cm, maximum = 173 cm TL) at the time of tagging. Throughout the study period, 111 (48.3% from all tagged individuals, 56 females and 55 males, Appendix A) bull sharks were tracked between 1 day and 9.4 years, during which the average number of days tracked was 1014 days. There was also no difference in the average number of days detected by size or sex (Appendix A).

Efficacy of tagging methods were compared by evaluating the frequency and longevity of detections of the internally vs. externally tagged bull sharks. Of the 233 sharks tagged, the 90 individuals who were externally tagged initially had a high frequency of redetection (up to 60% in 2018) but this decreased with time (to around 20%) and they were tracked for an average of 0.6 years (±0.08 years, range < 1 day to 3.1 years, Appendix A). In contrast, the 143 internally tagged sharks were tracked for an average of 3.9 years (±0.3 years, range < 3.3 days to 9.4 years, Appendix A), which was significantly different from the externally tagged sharks (ANOVA, *p*-value < 0.001), and they had a relatively stable redetection rate over time (around 30%, Appendix A). No differences were found among sexes and tag position (male external = 0.9 ± 0.1 years, female external = 0.5 ± 0.1 years; male internal = 3.9 ± 0.3 years, female internal = 4.0 ± 0.4 years). Furthermore, although bull sharks were first externally tagged in 2017, compared to 2009 for internally tagged sharks (Appendix A), preliminary data suggest that fewer externally tagged sharks were detected by other listening stations after their last detection on a shark listening station than internally tagged sharks (Appendix A).

### 3.1. Patterns of Bull Shark Occurrence

Sharks were tracked between 13 July 2017 and 9 January 2023. Individuals were detected between 2 and 1445 times by each shark listening station, and across up to 16 different locations (minimum = 1, mean = 5) (Figure 1B and Appendix A). The overall occupancy was greatest on the northern coast of NSW between Lennox Head and Forster (Figure 1A,B) with the largest number of sharks detected at Ballina Lighthouse (*n* = 53) and Ballina (*n* = 52) (Figure 2) and the greatest number of detections recorded at South West Rocks (*n* = 4259), followed by Yamba (*n* = 1761) and Forster (*n* = 1631). All three of these locations are within 800 m (minimum = 480 m, mean = 640 m, maximum = 770 m) of the nearest river mouth (Figure 1B).

There was no correlation between tagging location and the number of sharks detected by receivers adjacent to that location. For example, only four sharks were tagged adjacent to the South West Rocks shark listening station, yet this location represented the highest number of detections (Figure 2). Similarly, the total number of detections did not directly correspond to the number of sharks detected. Lighthouse Beach, Ballina recorded the highest number of individual sharks detected; however, this location was the fourth highest in terms of total number of detections (Figure 2).

An analysis of the detections of tagged juvenile sharks highlights the lack of detections by shark listening stations until they were 157 cm TL, with subsequent detections initially recorded at the listening stations near their natal river, followed by long-distance, latitudinal movements once they increased in size (Figure 3A–E). For example, a neonate, 80 cm TL bull shark, tagged inside the Clarence River on 6 December 2010, was only detected along the coastal beaches eight years post tagging when it reached an estimated size of 183 cm (Figure 3B). Larger juveniles (≥145 cm) tagged within the Clarence and Richmond Rivers had a greater frequency of detections by coastal stations within a year or two of tagging and were more frequently detected within 50 km of their tagging location before extending their range (Figure 3C–E).

### 3.2. Drivers of Bull Shark Occurrence

The detection data pooled across each biological class, the 5.5-year study period and all locations indicated that bull sharks were detected in lower latitudes waters almost year-round and there was a clear seasonality, with occurrences peaking in the southern latitudes in the austral summer through to late autumn/early winter (January–June) (Appendix A). The group-specific GAMs indicated significant effects for the interaction between latitude and year and binned hour of the day for juvenile and large bull sharks (Table 2). The occurrence of juvenile bull sharks peaked in northern NSW (<32° S) and decreased after 2020, because many individuals became >175 cm (Figure 4A). Similarly, large bull sharks were more frequently detected across the northern half of the state (Figure 4B); however, some sex-based segregation occurred with males being more prominent in northern NSW, while females exhibited a broader latitudinal spread in their detections (Appendix A). Intra-specific diurnal patterns of occurrence were observed, with juveniles exhibiting higher nearshore presence between 20:00 and 03:00 (Figure 4A), whilst the occurrence of large sharks gradually increased throughout the afternoon from midday through to 04:00, peaking around midnight (Figure 4B).

The environmental GAMMs (Table 3, Appendix A) revealed that water temperature was the primary variable influencing the presence of large bull sharks, with higher shark occurrence when temperatures were higher than 20 °C, with occurrence peaking at 23 °C (Figure 5A). Similarly, rainfall significantly influenced the occurrence of large bull sharks, with higher occurrence in nearshore areas after >45 mm of rain (Figure 5B). Occurrences were also greater when swell heights were >1.8 and <2.8 m (Figure 5C). However, there was no influence of moon phase nor tide on the nearshore presence of large bull sharks (Appendix A).

## 4. Discussion

Knowledge of the patterns of occurrence and the biological and environmental drivers of bull sharks in nearshore environments is essential for increasing the possibility for better predicting the likelihood of encountering bull sharks in the coastal areas of NSW. Using over 5.5 years of acoustic telemetry data from 111 bull sharks, we have shown the influence that environmental variables have on the occurrence and seasonal trends of bull sharks along the coast of NSW. Latitude was consistently the strongest predictor of occurrence over time, with similarities observed between juvenile and large sharks exhibiting overall highest occupancy in the coastal areas of northern NSW (<32° S) and within 800 m of a river. Significant associations with rivers were observed across all sizes; however, juvenile bull sharks were not usually detected by coastal shark listening stations until they were 157 cm total length. These size-based results support previous studies on the distribution patterns of juvenile bull sharks [37,51,52,53,54,55,56,57] with individuals remaining in their natal river for up to five years [56] or until 160 to 180 cm TL [58] due to the suggested increased availability of prey [59] and lower associated predation risk [60] compared to coastal environments [61]. Beyond this size, individuals transitioned from estuaries to marine areas [36] and this is reflected in changes in their diets and dentition [48,62]. Of those individuals that were <175 cm and that were detected in coastal areas, detections occurred only at night (20:00 to 03:00) and were confined to latitudes lower than 32° S throughout the year. The most plausible explanation for their association with lower latitudes is that juveniles prefer the warmer water temperatures that are experienced in these areas year-round. However, as detections of juveniles only constituted 15% of the overall data, and this size category is unlikely to be implicated in shark bites [3] due to their small mouth size [63], environmental factors influencing the nearshore occurrence of bull sharks were investigated only for large sharks >175 cm TL.

The distance to the nearest river also influenced the occurrence for large sharks, and was skewed towards lower latitudes, particularly for females. This may be related to their use of river systems as pupping grounds [64]. In the northern Atlantic, female bull sharks have expanded their distribution, making use of climate-change-induced warmer temperatures to pup in new estuaries in higher latitudes [65]. Along the east coast of Australia, the distribution of bull sharks has shifted poleward in the last decade, where many estuaries are available for potential pupping requirements [19]. These latitudinal patterns in distribution were consistent over the 5.5-year period and, albeit in such a relatively short time-frame, this indicates that bull shark distribution is exhibiting slight polar shifts in NSW nearshore waters. Our data, therefore, imply that bull shark use of these new potential pupping grounds can be expected with the projected ocean warming associated with the strengthening East Australian Current [66,67].

Individuals were also recorded revisiting coastal areas on multiple years post-release. Many animals, across a variety of taxa exhibit philopatry [68], with individuals repeatedly returning to the same stretch of coast for reproductive needs [66,69,70], foraging opportunities [37,71,72,73] or for reasons still largely unknown [26]. Additionally, ocean currents [19], water temperature [18,74,75,76,77], rainfall [38,39] and swell height [26,78] are well documented drivers of shark movement patterns and drivers of the behaviour shown here. Similarities among the broad-scale movement patterns of large bull sharks were observed, with large sharks exhibiting latitudinal migrations with increased detection during the austral summer and autumn. These broad-scale patterns of migrations are consistent with earlier studies [18,19,20,37,40,41], highlighting the influence that water temperature has on the connectivity between tropical and temperate regions in eastern Australia during the austral summer and autumn. Yet, here we document the southern-most record of bull sharks for the east coast of Australia, with a large male shark being detected off the coast of Merimbula (−36.9° S) during the austral autumn. The East Australian Current brings warmer water closer to shore during summer and autumn. These changing nearshore waters along the east coast of Australia have recently been shown to be contributing to changes in tiger shark [23] and bull shark [19] distribution. This range extension provides further evidence for the rapidly changing scenario being experienced in nearshore waters subjected to the influence of one of the fastest changing western boundary currents in the world [66,67].

Although this expansion into higher latitudinal nearshore areas may be a cause for concern due to the potential increased risk of negative shark interactions with water-users further south, this appears to be mitigated, to some extent, by the diel pattern of the occurrence of large bull sharks. Time of day was a key predictor of the occurrence of bull sharks. Similar to the diel patterns observed for juveniles, large bull sharks also frequent nearshore waters more at night, with an increased probability of nearshore presence in the afternoons and at night, with a peak occurrence between 18:00 and 01:00. The catch rates and detections of bull sharks at other localities support this diel pattern in the nearshore presence of sharks [14,18,38,43]. This late afternoon nearshore presence and early morning absence of bull sharks corroborates the shark bite patterns exhibited in KwaZulu-Natal, where this species is considered the primary shark responsible for interactions [79]; however, the potential risk for NSW cannot be calculated due to a paucity of data on surfer behaviour. The afternoon onshore winds during the austral summer and the reduced quality of waves likely leads to fewer surfers during this period of potential increased risk. Similarly, the nearshore presence of bull sharks is reduced in the mornings when surfers are more likely to be active. However, the presence of large bull sharks increased at swell heights between 1.8 m and 2.8 m, which likely increases the number of surfers using nearshore areas and, therefore, may represent conditions of increased potential risk. This risk would be accentuated during periods of high rainfall as this environmental variable was the second-most influential in predicting the occurrence of large sharks. Although rainfall data for the current study were localised in nature, rather than the catchment-level rainfall reported in Niella et al. [38], the increased detections following rainfall events imply sufficient run-off to influence animal behaviour and are consistent with studies published elsewhere [39].

The ability to carry out the long-term tracking of sharks provides a unique insight into the factors affecting their distribution, occurrence and drivers of movement. This is particularly pertinent during these times of rapidly changing coastal boundary currents and human-induced habitat alteration. The use of acoustic tags with 10-year lifespans potentially enables the assessment of individual animal adaptations to this ever-changing seascape; however, many telemetry studies are deploying tags externally into the musculature of the sharks [26,80]. Previous studies suggest bull sharks are particularly prone to prematurely shedding externally attached types of tags [81,82]. Dicken et al. [83] suggested that the high rate of tag shedding in adult sharks may be due to the extensive muscle mass at the base of the dorsal fin precluding piercing through the pterygiophores extending ventrally from the dorsal fin. A comparison of the detections of internally and externally tagged bull sharks highlights the difference in long-term tag detections between sharks carrying internal versus external tags. Although most internal tags were deployed before the externally tagged sharks (2–14 years versus <6 years, respectively), potentially reducing the ability for long-term comparison of tag retention, there appears to be a propensity for some bull sharks to shed external tags within months of tagging. Further research using multiple types of tags, on the same individual over the same periods of time, are required to assist long-term research into the retention of tags on bull sharks and shark responses to human-induced environmental impacts, whilst also contributing to the ongoing efforts in mitigating shark–human interactions.

### Implications for Management of Shark–Human Interactions

Using the largest dataset of acoustically tagged bull sharks in the world, we analysed the nearshore occurrence of bull sharks over an ~1000 km stretch of coastline that incorporates areas of high ocean water use and that has historically experienced shark bites [5] and community concern [29] throughout the region. The results from this study highlight the fact that the highest frequency of occurrence of large bull sharks in the coastal waters of NSW is found when the water temperature is between 22 and 24 °C, coinciding with the austral summer and autumn and between 18:00 and 01:00. We show that current shark bite mitigation educational messaging, incorporating proximity to rivers, turbidity (rainfall) and time of day, reflect the heightened periods of occurrence of large bull sharks. Our results suggest that time of day messaging for large bull shark presence should be modified from dawn and dusk to instead refer to afternoon and low-light periods. We concur with the current shark smart advice that nocturnal swimming and surfing, especially in warm waters and when water visibility is poor, should be avoided for many reasons, not the least of which being the presence of bull sharks.

## Figures and Tables

**Figure 1 biology-12-01189-f001:**
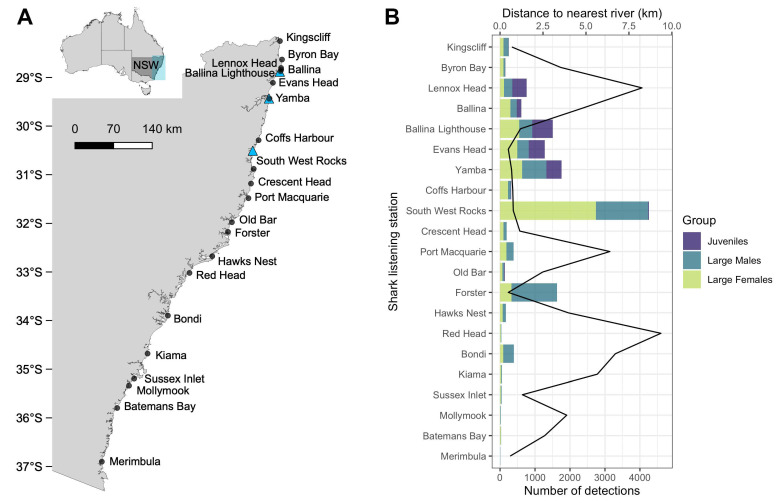
(**A**) Map of the coast of New South Wales (NSW) showing the locations of the 21 shark listening stations (points) and rivers (triangles) where juvenile bull sharks were tagged (rivers north to south: Richmond, *n* = 2, Clarence, *n* = 22, Bellinger, *n* = 5), and (**B**) total bull shark detections (bars) per biological group for each shark listening station from 2017 to 2023 and their respective distances to nearest river mouth (line).

**Figure 2 biology-12-01189-f002:**
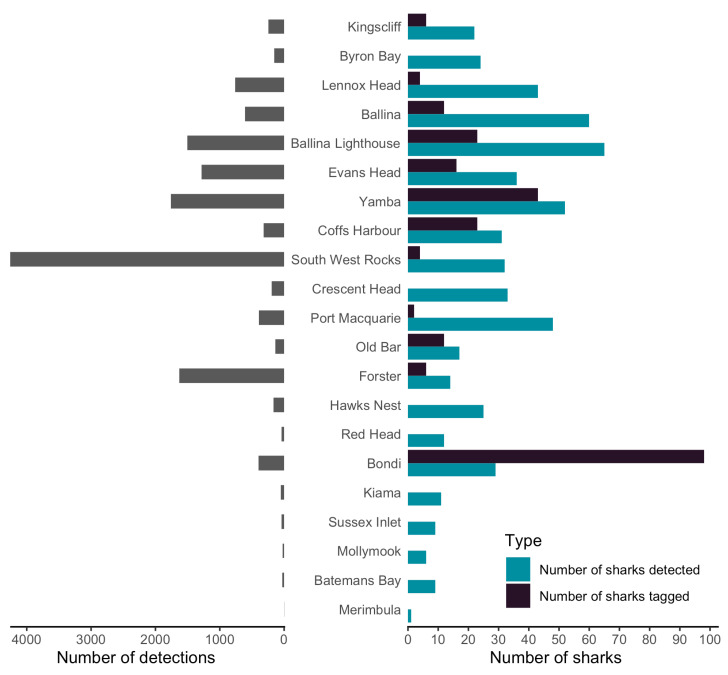
Summary of total detections (*n* = 13,996) and tagged vs. detected bull sharks by shark listening station location, including numbers of (1) total bull shark detections (*n* = 4044, after hourly binning), (2) tagged bull sharks (*n* = 233) between 4 March 2009 and 14 December 2022 and (3) individual sharks (*n* = 111) detected by receiver locations between 13 July 2017 and 8 January 2023.

**Figure 3 biology-12-01189-f003:**
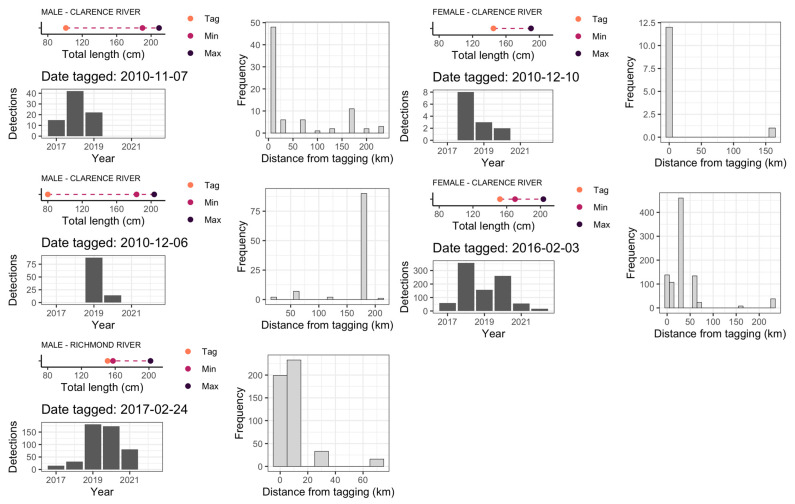
Male (left panel) and female (right panel) juvenile bull sharks (**A**–**E**) tagged inside the Clarence and Richmond Rivers and subsequently detected along the coast of New South Wales. Points represent respective shark sizes at tagging (Tag), and at first (Min) and last (Max) shark listening station detections. Histograms of detections by year, and as a function of distance from tagging locations, are included for each shark.

**Figure 4 biology-12-01189-f004:**
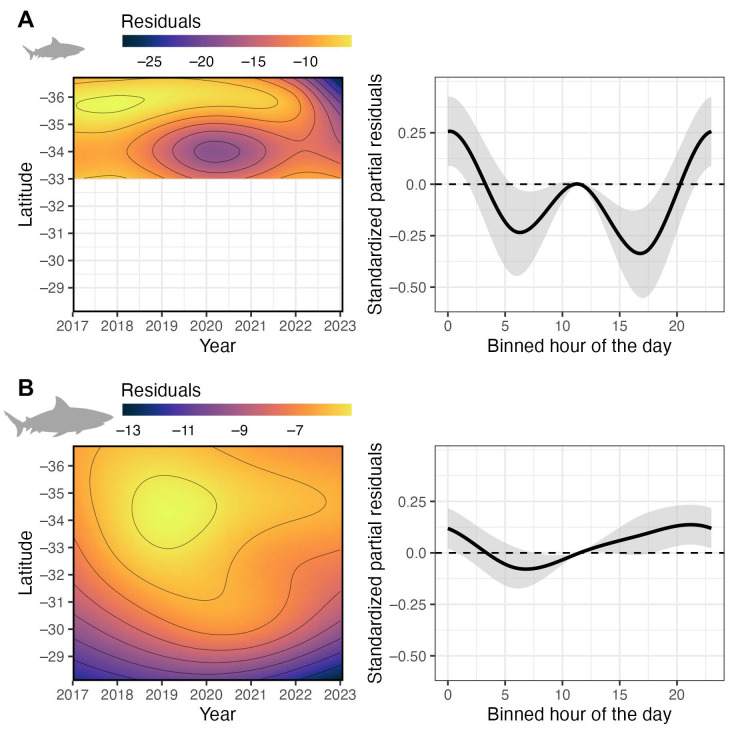
Group-specific spatio-temporal generalised additive models of (**A**) juvenile and (**B**) large bull shark occurrence along the coast of New South Wales as a function of the interacting effects between latitude degree and year (left panel) and binned hour of the day (right panel). The colour scales (left panel) represent the corresponding fitted model residuals. Horizontal dashed lines and shaded areas (right panel), respectively, represent the null effects and the 95% confidence intervals. Positive values on the vertical axis indicate an increased probability of occurrence, while negative values indicate an increased probability of absence (right panel).

**Figure 5 biology-12-01189-f005:**
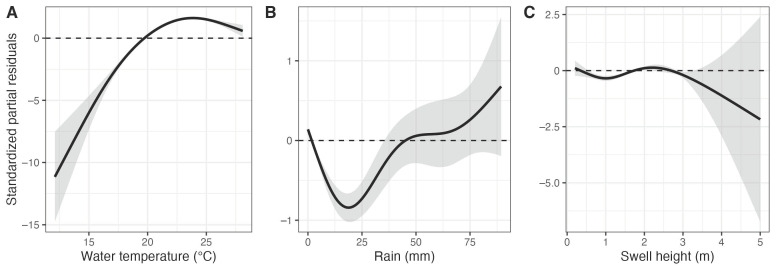
Response curves of environmental generalised additive mixed models predicting the occurrence of large bull sharks along the coast of New South Wales, including the significant effects of (**A**) water temperature, (**B**) daily rainfall and (**C**) swell height. Horizontal dashed lines and shaded areas, respectively, represent the null effects and the 95% confidence intervals, respectively. Positive values on the vertical axis indicate an increased probability of occurrence, while negative values indicate an increased probability of absence.

**Table 1 biology-12-01189-t001:** List of fixed candidate spatio-temporal and environmental explanatory variables tested in the generalised additive mixed models of the occurrence of bull sharks along the coast of NSW. Included are the respective units of measure, source and spline-based techniques used in the smoothing functions.

Group	Variable	Source	Spline
Spatio-temporal	Latitude (°)	Global Positioning System	Cubic-regression
	Binned hour of the day—Hour (h)	AEST/AEDT	Cyclic-cubic-regression
	Month	Calendar	Cyclic-cubic-regression
Environmental	Daily rainfall—Rain (mm)	Australian Bureau of Meteorology	Cubic-regression
	Lunar phase—Moon	R package ‘lunar’	Cyclic-cubic-regression
	Swell height—Height (m)	Manly Hydraulics Laboratory, NSW, Australia	Cubic-regression
	Tidal height—Tide (m)	Manly Hydraulics Laboratory, NSW, Australia	Cubic-regression
	Water temperature—Temperature (°C)	Sentinel tags	Cubic-regression

**Table 2 biology-12-01189-t002:** Generalised additive model results of juvenile (deviance explained = 16.3%) and adult (deviance explained = 5.2%) bull shark occurrence along the coast of NSW comprising the significant spatio-temporal interacting effects of latitude with year and binned hour of the day (Hour). Included are the effective degrees of freedom (Edf.), reference degrees of freedom (Ref.df.), Chi-squared (χ^2^) and *p*-value (*p*) of each model variable.

Group	Variable	Edf.	Ref.df.	χ^2^	*p*
Juvenile	Latitude × Year	13.76	13.98	1096.77	<0.001
	Hour	2.82	3.00	28.76	<0.001
Adult	Latitude × Year	13.35	13.90	1295.95	<0.001
	Hour	2.29	3.00	17.57	<0.001

**Table 3 biology-12-01189-t003:** Generalised additive mixed model (deviance explained = 8.4%) of the occurrence of large bull sharks along the coast of NSW, comprising their respective significant effects of environmental variables and year as a random effect. Included are the effective degrees of freedom (Edf.), reference degrees of freedom (Ref.df.), Chi-squared (χ^2^) and *p*-value (*p*) of each model variable.

Variable	Type	Edf.	Ref.df.	χ^2^	*p*
Temp	Fixed	3.71	3.92	1804.69	<0.001
Rain	Fixed	3.69	3.94	94.47	<0.001
Height	Fixed	3.16	3.45	38.96	<0.001
Year	Random	<0.01	1.00	<0.01	0.406

## Data Availability

The datasets used and/or analysed during the current study are available from the corresponding author on reasonable request.

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
