# Peer review of "Bull Shark (Carcharhinus leucas) Occurrence along Beaches of South-Eastern Australia: Understanding Where, When and Why"

_biology, 2023, doi:10.3390/biology12091189_

Round 1

Reviewer 1 Report

Dear Author/Authors,

This article discusses the occurrence and presence of bull sharks (Carcharhinus leucas) along the beaches of southeastern Australia. Bull sharks are one of the most dangerous sharks in the world and can be found in both saltwater and freshwater environments, making them a concern for swimmers, surfers, and boaters alike. While bull sharks are known to inhabit many regions of the world, this article focuses on their presence in the southeastern Australian coastline. Understanding the patterns of bull shark occurrence in these areas can help inform safety protocols and reduce the risk of shark-human interactions.

In the references, there are a total of 4 publications belonging to the article authors. Two of their own references in the article were made in 2016, 1 in 2021, and 1 in 2019. However, it would be more appropriate to review and reduce the citations made at the end of the text, especially those from the last 5 years.

It was stated that there were some missing references and some references were missing in the content of the text and in the reference section of the article, and they need to be corrected again.

Best Regards.

Author Response

Comment: In the references, there are a total of 4 publications belonging to the article authors. Two of their own references in the article were made in 2016, 1 in 2021, and 1 in 2019. However, it would be more appropriate to review and reduce the citations made at the end of the text, especially those from the last 5 years. 

Response: We are grateful to the reviewer for their review and acknowledge their point made about self-citations.  However, Smoothey et al. 2016 and 2019 are locally relevant citations and the first to report on the occurrence and drivers, respectively of bull sharks in this region.  For this reason, we believe they are relevant and have not been removed.

Comment: It was stated that there were some missing references and some references were missing in the content of the text and in the reference section of the article, and they need to be corrected again.

Response: We have amended the citations and references throughout the manuscript to address this point.

Reviewer 2 Report

This is a well written and readable MS about of Bull shark (Carcharhinus leucas) occurrence along beaches of south-eastern Australia. The informations that are provided, are important for both, to understanding the ecology of species and to improvement the measures in order to mitigation the human and sharks interaction.
I think that the MS should be published  in the journal after, however, some editorial corrections (eg. some figures (3 and 4) create  blunks on the  pages as well as the  table 2 are shared between two pages while its lenght doesn't justify this  fragmentation).

Author Response

Comments from Reviewer 2

This is a well written and readable MS about of Bull shark (Carcharhinus leucas) occurrence along beaches of south-eastern Australia. The informations that are provided, are important for both, to understanding the ecology of species and to improvement the measures in order to mitigation the human and sharks interaction.

Comment: I think that the MS should be published in the journal after, however, some editorial corrections (eg. some figures (3 and 4) create blunks on the pages as well as the table 2 are shared between two pages while its lenght doesn't justify this fragmentation).

Response: We thank the reviewer for their review and kind words.  We have amended the formatting of the figures and tables to ensure that they fit properly on the page and will ensure that when we receive the publication proofs that formatting issues are resolved.

Reviewer 3 Report

Comments

1.      The manuscript analyses an impressive set of acoustic tagging data on bull sharks covering almost the entire coastline of New South Wales, where fatality and severe injury to humans occur from interactions with bull sharks in coastal waters. The results of the analyses, integrating available environmental data, provide a valuable basis for improving management strategies for minimising human-bull shark interactions. 

2.      The manuscript’s subject matter is suitable for the journal Biology, and the manuscript is a valuable contribution to the literature. Therefore, I recommend its publication following minor revision based on the comments below.

3.      In lines 227‒229 and Table S1, there is mention of detections of acoustic transmitters at IMOS listening stations. However, no mention of their locations or how they integrate with the 21 stations described in the manuscript exists. From Table S1, there are more IMOS stations than project stations.

4.      In line 253, “binomial family of errors distribution” needs modifying. I suspect the authors mean ‘variation in the data, expressed as error, was explored using a family of binomial probability distribution functions’.

5.      In line 289, I suggest altering “observed between sexes” to ‘determined among sexes and tag position’.

6.      In Table S1, line 1, alter the expression “data format” to ‘date format’, and I suggest avoiding abbreviating ‘day/month/year’.

7.      In Table S1, I suggest altering the column heading “Location caught” to ‘Latitude tagged’.

The writing is generally clear. I do have a couple of editorial suggestions.

1.The subordinate clause should be separated from the main clause using a semicolon rather than a comma in lines 320 (i.e., “… sharks detected, however, this location …” should be ‘… sharks detected; however, this location …’). Similarly, in lines 347, 395, 464, and 483‒484, alter “…, however” to ‘…; however, …’.

2. It is usual to use an en-dash rather than a hyphen when referring to a range. For example, in line 165, “3-6 cm” should be ‘3‒6 cm’. See also lines 19, 42, 132, 150 (x2), 178, 302, 326, 332, 335, 342, and 372.

Author Response

Comments from Reviewer 3

  1. The manuscript analyses an impressive set of acoustic tagging data on bull sharks covering almost the entire coastline of New South Wales, where fatality and severe injury to humans occur from interactions with bull sharks in coastal waters. The results of the analyses, integrating available environmental data, provide a valuable basis for improving management strategies for minimising human-bull shark interactions. 
  2. The manuscript’s subject matter is suitable for the journal Biology, and the manuscript is a valuable contribution to the literature. Therefore, I recommend its publication following minor revision based on the comments below.

Comment: In lines 227‒229 and Table S1, there is mention of detections of acoustic transmitters at IMOS listening stations. However, no mention of their locations or how they integrate with the 21 stations described in the manuscript exists. From Table S1, there are more IMOS stations than project stations.

Response: Firstly, thank you to the reviewer for a detailed review of our manuscript.  In regards to the use of IMOS acoustic receivers, these stations were only used to look for detections of tagged sharks after the last detection on a shark listening station. The IMOS receivers were not used in the analyses and therefore, their locations were not described in the Material and Methods section of the paper.

Comment: In line 253, “binomial family of errors distribution” needs modifying. I suspect the authors mean ‘variation in the data, expressed as error, was explored using a family of binomial probability distribution functions’.

Response: This has been amended (L236) to “Variation in the data, expressed as error, was explored using a family of binomial probability distribution functions (Table 1).”

Comment: In line 289, I suggest altering “observed between sexes” to ‘determined among sexes and tag position’.

Response: This has been modified as suggested to (L272) “No differences were determined among sexes and tag position (male external = 0.9 ± 0.1 years, female external = 0.5 ± 0.1 years; male internal = 3.9 ± 0.3 years, female internal = 4.0 ± 0.4 years).”

Comment: In Table S1, line 1, alter the expression “data format” to ‘date format’, and I suggest avoiding abbreviating ‘day/month/year’.

Response: Table S1, line 1 has been amended to “day/month/year”

Comments on the Quality of English Language

The writing is generally clear. I do have a couple of editorial suggestions.

Comment: The subordinate clause should be separated from the main clause using a semicolon rather than a comma in lines 320 (i.e., “… sharks detected, however, this location …” should be ‘… sharks detected; however, this location …’). Similarly, in lines 347, 395, 464, and 483‒484, alter “…, however” to ‘…; however, …’.

Response: Amended throughout the manuscript.

Comment: It is usual to use an en-dash rather than a hyphen when referring to a range. For example, in line 165, “3-6 cm” should be ‘3‒6 cm’. See also lines 19, 42, 132, 150 (x2), 178, 302, 326, 332, 335, 342, and 372.

Response: Amended throughout the manuscript.